# A Review of CYP-Mediated Drug Interactions: Mechanisms and In Vitro Drug-Drug Interaction Assessment

**DOI:** 10.3390/biom14010099

**Published:** 2024-01-12

**Authors:** Jonghwa Lee, Jessica L. Beers, Raeanne M. Geffert, Klarissa D. Jackson

**Affiliations:** Division of Pharmacotherapy and Experimental Therapeutics, UNC Eshelman School of Pharmacy, University of North Carolina at Chapel Hill, Chapel Hill, NC 27599, USA; jlbeers@uw.edu (J.L.B.); geffert@unc.edu (R.M.G.)

**Keywords:** cytochrome P450, drug metabolism, drug-drug interaction, inhibition, induction, reaction phenotyping

## Abstract

Drug metabolism is a major determinant of drug concentrations in the body. Drug-drug interactions (DDIs) caused by the co-administration of multiple drugs can lead to alteration in the exposure of the victim drug, raising safety or effectiveness concerns. Assessment of the DDI potential starts with in vitro experiments to determine kinetic parameters and identify risks associated with the use of comedication that can inform future clinical studies. The diverse range of experimental models and techniques has significantly contributed to the examination of potential DDIs. Cytochrome P450 (CYP) enzymes are responsible for the biotransformation of many drugs on the market, making them frequently implicated in drug metabolism and DDIs. Consequently, there has been a growing focus on the assessment of DDI risk for CYPs. This review article provides mechanistic insights underlying CYP inhibition/induction and an overview of the in vitro assessment of CYP-mediated DDIs.

## 1. Introduction

In clinical practice, the increasing prevalence of multiple drug therapy presents an ongoing challenge, with drug interactions being a significant concern [1]. These interactions, known as drug-drug interactions (DDIs), often result from the changes in a victim drug’s plasma concentrations due to a perpetrator drug either inhibiting or inducing the metabolism or transport of the victim drug [1,2]. Consequently, DDIs may lead to changes in the pharmacokinetic (PK) profile with reduced efficacy and/or unexpected toxicities. Enzyme- or transporter-mediated inhibition of drug elimination can lead to increased area under the curve (AUC), maximum plasma concentration (C_max_), and half-life (t_1/2_) [3]. Conversely, induction generally leads to reduced AUC, C_max_, and t_1/2_. Several drugs, such as terfenadine, mibefradil, cisapride, and nefazodone, have been removed from the market due to adverse reactions mediated by DDIs, necessitating assessment of DDI potential during drug development and post-marketing approval [1,4]. Assessment of the potential for DDI of a drug comprises three steps [5]. First, it requires the identification of the primary pathways through which the drug is eliminated from the body. Second, it involves estimating the contribution of drug-metabolizing enzymes and transporters to the disposition in the body. Lastly, it entails characterizing how the drug influences the function of drug-metabolizing enzymes and transporters.

Many therapeutic drugs undergo hepatic metabolism. Thus, when evaluating DDIs, PK interaction studies focus on the association of the test drug with drug-metabolizing enzymes [2,6]. Cytochrome P450 (CYP) enzymes are a superfamily of enzymes primarily found in the liver, but also in other tissues including the intestines, kidneys, lungs, and brain. CYPs are capable of catalyzing oxidative biotransformation of 70–80% of drugs in the market [7]. Among the 57 putatively functional human CYPs, CYP1A2, CYP2B6, CYP2C8, CYP2C9, CYP2C19, CYP2D6, and CYP3A4/5 are the predominant forms expressed in the liver, and they are frequently implicated in drug metabolism and DDIs [2,7]. Regulatory agencies frequently update their guidelines on DDI studies, with particular focus on CYP enzymes, due to safety concerns associated with these interactions. The guidelines provide considerations when choosing experimental conditions such as test system, probe substrates, and positive controls [2].

The United States Food and Drug Administration (FDA) recently issued a new guideline standard entitled “In vitro Drug Interaction Studies-Cytochrome P450 Enzyme- and Transporter-Mediated Drug Interactions” in 2020. Results from in vitro experiments can be translated for in vivo predictions or the design of clinical studies by using various modeling methods [2,5]. Examples of commonly used in vitro marker reactions, inhibitors, and inducers for CYP-mediated drug metabolism are shown in Table 1. This article reviews the current understanding of mechanisms of CYP inhibition/induction and in vitro approaches to assess CYP-mediated DDIs. It is important to acknowledge that while CYPs are often associated with drug-metabolizing enzymes and may be the main route of elimination for many drugs, transporters also contribute to the appearance of DDIs and should also be characterized according to the FDA guidance [5,8]. However, transporter-mediated DDIs are not in the scope of this review.

## 2. Reaction Phenotyping

### 2.1. Overview of Reaction Phenotyping

It is critical to determine if a drug candidate is a substrate for a CYP enzyme prior to administration to a patient. This is to ensure the drug does not have DDIs that may be of clinical significance. Additionally, knowledge of major metabolizing enzymes can help with any pharmacogenetic, disease state, or environmental considerations [68]. Reaction phenotyping is a commonly used in vitro approach to identify the enzymes and pathways responsible for metabolizing a drug [69,70,71,72]. This is an important step to assess the potential for DDIs as some metabolic pathways may compete for the same enzymes. This type of characterization is recommended for industry sponsors according to the FDA guidance [5]. Understanding the in vitro contributions of CYP enzymes to the overall metabolism of a drug is one of the first steps to determining the need for clinical DDI studies. The main goals of reaction phenotyping are to (1) determine the fraction metabolized by each CYP involved in the metabolic clearance, (2) characterize enzyme kinetics and other in vitro parameters, and (3) provide an early screen for potential DDIs [68,69,73,74,75].

#### 2.1.1. Fraction Metabolized (f_m_)

The fraction metabolized (f_m_) refers to the extent a substrate undergoes hepatic metabolism by a specific enzyme. The f_m_ value is specific to an individual enzyme and substrate. A high f_m_ value (f_m_ > 0.9) means one enzyme is primarily responsible for the majority of the metabolism of a substrate. This is considered a DDI concern especially if the main route of elimination for the compound is metabolism. In general, the pharmaceutical industry seeks to reduce the f_m_ value of a compound by introducing structural modifications to reduce the risk for the drug candidate being a victim of DDIs [74]. The f_m_ can be determined by multiple reaction phenotyping approaches, which are explained in the following sections. The f_m_ value is described by:$$f_{m=\frac{{CL}_{int,sc,u\;{CYP}_{x}}}{{CL}_{int,sc,u\;{CYP}_{total}}}}$$
where CL_int,sc,u_ CYP_x_ represents the intersystem extrapolation factor (ISEF) scaled value from a single CYP enzyme, and the CL_int,sc,u_CYP_total_ term is the sum of all scaled values in the system.

#### 2.1.2. In Vitro Pharmacokinetic Parameters

Examining in vitro PK parameters is crucial for understanding in vivo effects [2,76]. At steady state, the concentration of the enzyme–substrate complex, treated as a reactive intermediate, remains constant, and any fluctuations in substrate concentration are insignificant [76]. This assumption underlies efforts to maintain minimal substrate turnover (<10%) in experiments aimed at determining enzyme kinetic parameters. It also implies a necessity for the substrate concentration to significantly exceed the enzyme concentration. Therefore, alterations in substrate concentration resulting from enzyme–substrate complex formation are deemed negligible, necessitating the maintenance of the lowest possible enzyme concentration to meet this criterion [76]. These parameters include the V_max_, or the maximal velocity (or rate) that an enzyme can catalyze a reaction when it is fully saturated with substrate; K_m_, the substrate concentration at half-maximal enzyme velocity; the CL_int_, the intrinsic clearance or efficiency that a process can eliminate a drug, specifically through metabolism; t_1/2_, or half-life, the time it takes for half of the drug to be eliminated; and f_CL_, or the fraction of metabolic clearance proceeding through a pathway [11]. The CL_int_,_u_ is calculated as follows:CLint,u=(VmaxKm × CYP abundance per mg of HLM)Fu,mic×ISEF(CLint)
where CL_int,u_ is the intrinsic unbound clearance, V_max_ is the maximal velocity, K_m_ is the Michaelis–Menten constant, F_u,mic_ is the fraction unbound in microsomes, HLM is human liver microsomes, and ISEF is the intersystem extrapolation factor (see below) [77]. It is important to determine the unbound value for clearance as the unbound drug is responsible for pharmacological activity in the system, and is able to cross membranes, whereas the bound drug is not. However, this is often difficult to determine as total drug concentration is typically sampled in these systems. For highly bound drugs, the FDA guidance states that the unbound fraction in plasma should be considered 1% (f_u,p_ = 0.01), if experimentally determined to be less than one percent. Additionally, the CL_int_ may be determined through substrate depletion experiments. These experiments may be completed with microsomes or hepatocytes in order to estimate in vitro intrinsic clearance [78].

#### 2.1.3. Drug-Drug Interaction Implications

When considering PK DDIs, the equation by Rowland and Matin is often used [79]:$$Ratio=\frac{1}{\frac{f_{m}}{1+\left(\frac{\left[I\right]}{K_{I}}\right)}+(1-f_{m})}$$
where f_m_ is the fraction metabolized, [I] is the inhibitor concentration, and K_I_ is the inhibition constant. For DDI predictions, the ratio is the AUC in the presence and absence of inhibitor (AUCR). Based on this equation, when f_m_ increases, the potential for a DDI also increases [79]. This shows the relationship between f_m_ value and DDI risk, and therefore highlights the importance of determining the f_m_ value. The FDA guidance recommends in vivo clinical studies when ≥25% of the clearance is from one enzyme [5]. However, definitive information for the major drug clearance pathways can only be determined from an in vivo radiolabeled study [74].

### 2.2. Reaction Phenotyping Approaches

There are three main approaches to reaction phenotyping: (1) selective inhibition with chemical inhibitors or antibodies, (2) an individually cDNA expressed recombinant CYP (rCYP) panel, and (3) correlation analyses to CYP activities determined in individual human liver microsome (HLM) donors [69]. These approaches may be used individually, but are often combined, and results are compared for accuracy and agreement [80,81].

#### 2.2.1. Chemical Inhibition Approach

The chemical inhibition approach refers to using chemical inhibitors with well-defined and specific selectivity in a human-derived in vitro hepatic system [75]. This is often performed with a single concentration of a well-studied inhibitor [2,68,75]. It is critical that the inhibitor has defined selectivity toward a specific target enzyme. Recommended chemical inhibitors are described by the FDA in vitro selective inhibitors for CYP-mediated metabolism and are included in Table 1 [82]. The potency and selectivity of inhibitors should be assessed prior to use in a reaction phenotyping study [2,68,75]. Inhibitory antibodies can also be used for this approach [83,84]. However, these antibodies do not always reach maximal inhibition [84].

#### 2.2.2. Recombinant CYP Panel (RAF/ISEF Method) Approach

In this method for reaction phenotyping, a panel of rCYP enzymes with known activity and normalized for protein are incubated with the drug of interest, and a scaling factor is applied to the final parameters to account for the full liver rather than only the individual enzymes [73,85]. The panel typically includes CYP1A2, 2B6, 2C8, 2C9, 2C19, 2D6, and 3A4 and 3A5 [5,80]. Once the in vitro kinetic parameters are scaled, the CL_int_ is estimated [81]. Either the ISEF or the relative activity factor (RAF) may be used. The ISEF scaling factor is calculated as follows:CLint ISEF=VmaxKm(HLM)VmaxKmrCYP × CYP abundance (HLM)
where CYP abundance (HLM) is represented as the picomoles of CYP per milligram of protein [73,85]. It is important to note that scaling factors can be affected by which probe substrate is used to derive them.

The RAF is calculated as follows [86]:RAF=Vmax,HLMKm,u,HLMVmax,rCYPKm,u,rCYP
where K_m,u_ is the K_m_ corrected for unbound fraction. These equations are used to scale the in vitro clearance values from the recombinant CYP enzyme to HLMs.

#### 2.2.3. Correlation Analysis Approach

The correlation analysis approach uses the relationship between the rate of a metabolite formed and the marker activity of a specific CYP in a panel of HLMs from multiple donors [2]. A major limitation of the correlation analysis is that it does not provide f_m_ values and is generally only used when the contribution by a single enzyme is significant [80].

#### 2.2.4. Qualitative-then-Quantitative Approach

A new methodology has recently been developed that sequentially combines the use of an rCYP panel followed by use of varying concentrations of selective chemical inhibitors in pooled HLM [75]. In this qualitative-then-quantitative approach, an rCYP screening panel is employed to qualitatively show which CYP enzymes have the ability to metabolize the parent drug to metabolites of interest. Following the qualitative screen, the drug is incubated in HLM with increasing concentrations of selective inhibitors for the CYP enzymes identified in the first step to quantify the extent of inhibition. Departing from using one or two of the previously described methodologies and comparing for agreement, this new approach adopts a pre-defined selection of CYPs that have been shown to metabolize the drug. The subsequent detailing approach with selective inhibitors is used to determine the fraction of metabolic clearance through a particular pathway (f_CL_) and ultimately, the f_m_. This approach yields more accurate estimations of f_m_ not overestimating contributions of CYP enzymes to the overall metabolism. Additionally, it addresses the issue that using two different methods of reaction phenotyping often results in f_m_ values that do not agree.

The qualitative-then-quantitative methodology has been applied in the literature and has revealed the metabolism of the antibiotic drug linezolid to be more complex and implicated additional CYP enzymes than previously reported [75,87]. The expanded rCYP screening by Wyndalda et al. included the most commonly implicated CYP enzymes (CYP1A2, 2C8, 2C9, 2C19, 2D6, 3A4, and 3A5) and less commonly identified CYPs in drug metabolism (CYP1A1, 2A6, 2B6, 2E1, and 4A11) [88]. In a recent study, the use of additional rCYPs showed that CYP2J2, 4F2, and 1B1 metabolize linezolid, which was not previously shown in the literature [87]. The specific contributions of each CYP were then determined with specific inhibitors of each CYP involved in metabolism. This application of the qualitative-then-quantitative approach shows how important enzymatic contributions may be overlooked by using only the chemical inhibition approach or a less-extensive rCYP panel [75,87].

#### 2.2.5. Additional Methodologies

There are additional methods that may be used in conjunction with the approaches defined previously. Radioactive ([^14^C]-labeled) compounds are often used and usually preferred to non-labeled compounds in in vitro systems. The assessment and calculation of f_m_ and RAF/ISEF scaling factors are most accurately accomplished with the use of a radiolabeled compound, which is typically available during pre-clinical development [81]. If non-radiolabeled compounds are used, the results should be followed up with a bioanalytical study [80].

Machine learning algorithms are being investigated to better predict in vitro to in vivo extrapolations (IVIVE) from in vitro work. It is challenging to determine in vivo clearance and f_m_ and new models may help to bridge the gap. Recent work has used only the structure of the compound to predict the contribution ratio of a specific enzyme involved in metabolism [89]. This work has shown that in vivo values can be predicted in this way and are similar to in vitro values obtained in the lab [89]. Other work in this space includes creating machine learning models to extrapolate in vivo parameters from in vitro inputs. Another study compared various machine learning models, which predicted in vivo human intravenous (IV) clearance from in vitro data, and showed that the machine learning models were able to predict the in vivo clearance values for 16 Pfizer compounds from their respective clinical studies [90]. This model used inputs from over 600 molecules and used chemical structures, ionization, and logP, and in vitro experimental parameters including CL_int_, cell density, and f_u_, in order to predict in vivo IV clearance. SIMCYP^®^ software (Certara, Sheffield, UK) has also been employed to calculate the fraction and rate of metabolism of compounds based on in vitro data. The in vitro data was then used to predict DDI issues with ketoconazole, which may be concomitantly administered [91]. Modeling frameworks based on in vitro data can help to guide researchers on how and when to proceed with clinical DDI studies and provide substantial information on disposition before first-in-human (FIH) trials.

### 2.3. Limitations of Reaction Phenotyping

Although reaction phenotyping is relatively quick, robust, and reproducible in validated systems, there are limitations of the commonly used systems that must be considered. A main concern of using the selective inhibitor approach is the lack of specificity of some inhibitors [58,92,93]. A proposed strategy to overcome this limitation is a six-parameter inhibition curve fitting approach [68]. This method can generate more accurate values for estimates of enzyme contributions by compensating for overlapping effects of inhibition profiles [73].

Another limitation is that of the HLM system. This system only maintains activity for about 1–2 h once thawed [94]. Hepatocytes in suspension can also be a viable option but only have a 4–6 h incubation period [95,96]. This provides a challenge for low turnover compounds, which may have a significantly longer half-life in vitro. Some work has been completed to provide other options for these compounds. The HepatoPac^®^ co-culture model has been used as a new model for low-turnover compounds [97,98]. The turnover of slowly metabolized compounds alprazolam and tolbutamide was two-fold greater in the HepatoPac^®^ model as compared to only suspended hepatocytes [97]. In a separate study, f_m_ values for 10 out of 13 CYP3A4 substrates were determined to be within two-fold of the in vivo value [98]. Validated alternative systems can be a good way to overcome the limitations of the HLM system.

These alternative systems can also help to overcome another limitation of the HLM system. The microsomal system is a fraction of the liver that contains enzymes localized in the endoplasmic reticulum membrane, including CYPs, flavin-containing monooxygenase (FMO), and UDP-glucuronyltrasferases (UGTs) [99,100]. This excludes cytosolic enzymes, including aldehyde oxidase (AO), monoamine oxidase (MAO), xanthine oxidase (XO), and alcohol/aldehyde dehydrogenase (ADH/ALDH) [5]. Carboxylesterases (CES) are localized in both the microsomal and cytosolic fractions. Although it is rare for a cytosolic enzyme to be the primary metabolizing enzyme for a drug, using a microsomal system alone may hide the contributions of cytosolic enzymes [101].

## 3. CYP Inhibition

### 3.1. Mechanisms of CYP Inhibition

Assessment of a drug’s potential to inhibit CYP enzymes is a multifaceted process that begins early in the preclinical phase of drug development. CYP enzymes typically contain both active and allosteric sites for binding multiple ligands, which may act as substrates, inhibitors, and/or activators. CYP inhibition may be broadly divided into reversible, quasi-irreversible, and irreversible inhibition.

#### 3.1.1. Reversible Inhibition

There are four types of reversible inhibition: competitive, non-competitive, uncompetitive, and mixed competitive/non-competitive inhibition [102]. For all types of reversible inhibition, enzyme function is restored after dissociation of the inhibitor from the active or allosteric site. The duration of reversible inhibition in vivo therefore depends on the elimination half-life of the inhibitor. Dissociation of a reversible inhibitor from an enzyme is described by the equilibrium dissociation constant K_i_. Equations describing each form of reversible inhibition are shown in Figure 1.

Generally considered the most common and well-understood form of inhibition, competitive inhibition occurs when two molecules “compete” for mutually exclusive, reversible binding at the same active site of a CYP enzyme, thereby reducing the amount of enzyme available to metabolize a drug when a competitive inhibitor is present [102,103]. Though the V_max_ of the reaction is unaffected, the K_m_ is increased as a result of competition for the same active site [104]. Non-competitive inhibition occurs when an inhibitor is capable of binding at an allosteric site, regardless of whether a substrate is bound in the active site [102]. Substrates may still bind to the active site following non-competitive binding; however, the resulting enzyme–substrate–inhibitor complex is inactive [103]. Since non-competitive inhibitors do not impact substrate binding, the K_m_ remains unchanged while the V_max_ decreases as a consequence of inhibition [102,104]. An uncompetitive inhibitor lacks affinity for the free enzyme and is only capable of binding at an allosteric site when the enzyme is bound to the substrate (i.e., the enzyme–substrate complex) [102,103]. Like non-competitive inhibition, the enzyme–substrate–inhibitor complex is rendered inactive [103]. Uncompetitive inhibitors reduce the V_max_ of the reaction by reducing the number of functional enzyme–substrate complexes, which leads to a corresponding decrease in K_m_ as the reaction seeks equilibrium [102]. Mixed competitive/non-competitive inhibition (commonly referred to as simply “mixed inhibition”) is a distinct type of non-competitive inhibition, in which the inhibitor binds to the allosteric site with varying affinity based on whether a substrate is bound in the active site by a factor α [102]. This necessitates two terms for inhibitor dissociation when describing mixed inhibition kinetics: K_i_ for dissociation of the inhibitor from the enzyme-inhibitor complex, and αK_i_ for dissociation of the inhibitor from the enzyme–substrate–inhibitor complex [102]. As a result, mixed inhibitors decrease the V_max_ of the reaction and may either increase or decrease the K_m_ depending on the value of α [102].

The type of reversible inhibition enacted by an inhibitor may be identified experimentally using classic Michaelis–Menten enzyme kinetic experiments for determining the V_max_ and the K_m_ with and without the inhibitor [102,104]. Using these results, graphing the inverse of the measured metabolite formation rate vs. the inverse of the substrate concentration generates a Lineweaver–Burk plot. Each type of reversible inhibition is associated with a characteristic shift in the resulting Michaelis–Menten and Lineweaver–Burk plots, shown in Figure 1. Historically, the enzyme kinetic constants and mechanism of inhibition were determined by visually examining the shift in slope (equal to K_m_/V_max_), x-intercept (equal to −1/K_m_), and y-intercept (equal to 1/V_max_) observed with an inhibitor on the Lineweaver–Burk plot [104]. Today, the most accurate and preferred method for determining kinetic constants is to use nonlinear regression to fit the Michaelis–Menten model directly to the non-transformed data [102]. The type of inhibition may then be identified by determining which reversible inhibition model best fits the experimental data (represented by the equations in Figure 1). These analyses may be performed using one of many widely accepted statistical software packages, such as GraphPad Prism.

Importantly, experiment conditions should be optimized prior to performing enzyme kinetic experiments for inhibition. To accurately measure kinetic parameters, the linearity of metabolite formation with respect to both incubation time and protein concentration must be established beforehand. These experiments should be performed using a substrate concentration approaching the K_m_. The shortest incubation time and lowest protein concentration within the linear range should be selected for performing kinetic experiments to minimize depletion of the substrate [102]. The concentration of inhibitor should approximate the in vivo concentration at the CYP active site [102].

The degree of reversible inhibition may be measured as a ratio of intrinsic clearance values for a probe CYP substrate in the presence and absence of the interacting drug. This ratio is referred to as the R_1_ value and is calculated as follows:$$R_{1}=1+\frac{I_{max,u}}{K_{i,u}}$$
where I_max,u_ is the maximal unbound plasma concentration of the interacting drug at steady state, and K_i_._u_ is the in vitro unbound inhibitor dissociation constant [5]. If R_1_ is ≥1.02, further investigation using predictive modeling techniques or a clinical DDI study is warranted [5,105].

Reversible inhibition may be differentiated from other forms of inhibition in vitro by determining the effect of preincubating the inhibitor with the CYP enzyme before adding the substrate to reactions. Irreversible inhibitors display time-dependent, saturable kinetics, and the length of the preincubation period correlates with the degree of inhibition [103]. In contrast, the degree of reversible inhibition is unaffected by preincubation time. Time-dependent inhibitors are thus defined by an observed increase in the extent of inhibition when a preincubation period is added to reactions [103,106]. Time-dependent and mechanism-based inhibition are closely related but separately defined terms. Whereas time-dependent inhibitors are identified by the kinetics of enzyme inhibition experimentally, mechanism-based inhibition is a type of time-dependent inhibition where the inhibitor binds to the active site and subsequently inactivates the enzyme [106].

To experimentally determine whether a compound is a time-dependent inhibitor, probe substrate assays are performed with the addition of a preincubation step [106,107]. This is typically accomplished using what is commonly known as the “dilution method”, in which the inhibitor is first preincubated with a high concentration of CYP enzyme(s) and NADPH. Aliquots of this incubation are then diluted into a second incubation containing the substrate and NADPH at specified time points. These incubations are performed using a range of inhibitor concentrations to determine the k_inact_ and K_i_ values [106]. The resulting rates of metabolite formation can then be compared to matched incubations without inhibitors to determine the degree of time-dependent inhibition [106,107]. The latest FDA guidance for industry on in vitro drug interaction studies recommends screening for both reversible and time-dependent inhibition of CYP1A2, CYP2B6, CYP2C8, CYP2C9, CYP2C19, CYP2D6, and CYP3A [5]. The decision tree presented in Figure 2 illustrates the process for evaluating CYP inhibition for new lead compounds.

#### 3.1.2. Quasi-Irreversible Inhibition

Some substrates are transformed by CYPs into metabolite intermediates that form stable, inactive complexes with the prosthetic heme group of the enzyme, known as a metabolite intermediate complex. Metabolite intermediate complexes are stable at physiologic conditions and can be detected experimentally using imaging spectroscopy [61,106,108]. This process is called quasi-irreversible inhibition because decomplexation can technically occur in the presence of lipophilic compounds that can displace the intermediate and restore activity in vitro [103,109]. Incubation with the oxidizing agent potassium ferricyanide can be performed to recover enzyme activity following quasi-irreversible inhibition [109,110]. In this context, decomplexation refers to displacement of the metabolic intermediate from the iron-containing coordination complex (i.e., the heme group) located in the active site of CYP enzymes [103,111].

#### 3.1.3. Irreversible Inhibition

Irreversible inhibition occurs when an inhibitor irreversibly inactivates a CYP enzyme, typically either through alkylation of the heme moiety or covalent binding to apoprotein (i.e., protein without a bound cofactor) [103]. Since this form of inhibition relies on metabolic activation, it is also commonly referred to as mechanism-based or suicide inhibition [103]. Importantly, so-called irreversible inhibition does not always result in complete loss of CYP enzyme activity. In some cases, covalent binding may be reversible, and activity may return over time [111]. Examples of reversible covalent inhibitors include the anticancer agent bortezomib, which reversibly inhibits the 26S proteasome by binding to a threonine residue in its catalytic site, and the antidiabetic agent saxagliptin, which covalently modifies a serine residue in the active site of dipeptyl peptidase-4 [112]. In other cases, the dissociation rate constant k_off_ for the substrate or the resulting metabolite is very small, causing de facto irreversible inhibition [103,113].

Restoration of enzyme activity following irreversible inhibition is typically based on the rate at which the affected tissue(s) synthesize new CYP proteins. The estimated recovery half-life for different CYP enzymes has ranged from 20 to 50 h depending on the individual enzyme affected and the elimination half-life of the inhibitor tested [114,115].

Like the R_1_ value for reversible inhibition, the R_2_ value for estimating the degree of time-dependent inhibition is determined as a ratio [105]. The R_2_ value is calculated using the natural in vivo enzyme degradation rate constant (k_deg_) [106] and the observed inactivation rate of the affected enzyme (k_obs_):$$R_{2}=\frac{{k_{obs}+k}_{deg}}{k_{deg}}$$K_obs_ may be calculated using the maximal inactivation rate constant (k_inact_), the unbound inhibitor concentration at half-maximal inactivation (K_I,u_), and maximal unbound plasma concentration of the interacting drug at steady state (I_max,u_) [5]:$$k_{obs}=\frac{{k_{inact}\times 50\times I}_{max,u}}{K_{I,u}+50\times I_{max,u}}$$

If R_2_ is ≥1.25, the FDA recommends further investigation of the drug’s interaction potential using model-based predictions or clinical DDI studies using a clinical index substrate [5].

### 3.2. Methods for Assessing CYP Inhibition

#### 3.2.1. Early High-Throughput Screening

Plate-based fluorescent and luminescent assays can be used in early high throughput screening to determine a drug’s inhibitory potential. These experiments typically involve dosing rCYPs in a 96-well format with a pro-fluorescent or pro-luminescent substrate that generates metabolites that can be detected using a plate reader. The IC_50_ value (the concentration of inhibitor resulting in half-maximal inhibition) may then be calculated by measuring the difference in signal with a range of inhibitor concentrations compared to control incubations [116,117,118]. While such assays offer the advantage of high throughput and sensitivity, they are typically only performed with rCYPs since these substrates are not selective for individual CYP enzymes [117,119]. Furthermore, the assay must reliably yield metabolites with an attached fluorophore to avoid non-specific fluorescence.

Cocktails containing multiple selective CYP substrates are also commonly used with microsomes to increase throughput in combination with LC-MS metabolite profiling to screen for inhibition of multiple CYPs at once [116,119]. The Basel cocktail, for example, has been validated for both in vitro and in vivo DDI studies, and contains the following CYP substrates: caffeine (CYP1A2), efavirenz (CYP2B6), losartan (CYP2C9), omeprazole (CYP2C19), metoprolol (CYP2D6), and midazolam (CYP3A) [120].

One additional method is the use of radiolabeled selective CYP substrates, which typically release radiolabeled water or formaldehyde when metabolized [116,117,121]. This approach requires solid phase extraction or scintillation proximity assay beads to isolate radioactive metabolites [116,117,121].

#### 3.2.2. Probe Assays for CYP Inhibition

Following initial screening, validated probe assays with HLM are commonly performed prior to and/or alongside phase I studies. A comprehensive list of both in vitro and clinical index substrates, inhibitors, and inducers for major CYP enzymes and transporters has been published by the U.S. FDA for DDI screening (see Table 1 for CYP enzymes) [82]. Importantly, these marker substrate reactions, while useful, are accompanied by several limitations. Although these studies are considered the industry standard for measuring drug inhibition, few (if any) substrates are fully selective for a single CYP enzyme. Often, multiple metabolites are formed simultaneously through multiple pathways, which may affect results if the inhibitor is also metabolized by one or more of the same enzymes as the substrate. Drug metabolites may also inhibit CYPs to an equal or greater extent than the parent drug [122]. Investigators should consider these factors beforehand and select an alternative CYP substrate for microsomal studies if needed. This issue may be partially circumvented by using individual rCYP enzymes, though this system comes with its own set of limitations. Kinetic parameters calculated using recombinant enzymes may show a lack of correlation to those calculated using microsomes due in part to over-expression of the enzyme and the lack of additional drug-metabolizing enzymes. Further validation with microsomes is recommended after initial screening with recombinant enzymes [123].

In addition, drug-metabolizing enzymes, particularly those in the CYP3A subfamily, may have flexible active sites that can be inhibited through different mechanisms depending on the structure of the bound substrate [124]. Inhibition of CYP3A enzymes should ideally be assessed using two different marker reactions (typically midazolam 1′-hydroxylation, testosterone 6β-hydroxylation, and/or felodipine dehydrogenation). Additional considerations for substrate selection include the rate of metabolite generation in vitro and the availability of metabolite standards and internal standards for LC-MS analysis [5,119,124].

For clinical DDI studies, an ideal index substrate is sensitive enough to demonstrate measurable changes in exposure when administered with an inhibitor. The AUC of a sensitive index substrate should increase by at least five-fold with a strong inhibitor, and moderately sensitive substrates should demonstrate an AUC increase of two to five-fold with a strong inhibitor [5].

#### 3.2.3. Model-Based Approaches for Predicting CYP Inhibition

The recent adoption of high throughput screening in DDI assessment has led to the use of predictive modeling to analyze large datasets for potential CYP-mediated DDIs. In silico prediction of CYP-mediated metabolism carries several advantages over traditional in vitro approaches: modeling may be performed very early in drug development to quickly assess many compounds at a low cost [125]. In addition, modeling may be performed with compounds that have yet to be synthesized [125].

Predictive models are commonly developed using a ligand- and/or structure-based approach and validated through cross-validation or with large external datasets of known substrates and inhibitors [125,126,127]. Following a ligand-based approach, large databases of structurally diverse compounds are screened for binding and inhibition of CYPs based on quantitative structure–activity relationships [125]. Structure-based models instead rely on three-dimensional CYP protein structures obtained through x-ray crystallography or NMR, and predictions are made based on docking simulations [125,128]. For both model types, multiple linear regression and/or machine learning techniques are used to make predictions [125].

## 4. CYP Induction

### 4.1. Mechanisms of CYP Induction

Enzyme induction refers to a mechanism characterized by increased expression and activity of an enzyme resulting from exposure to a xenobiotic or endogenous inducing agent [129]. The increased clearance of the drug results in lower concentrations and may lead to a diminished pharmacological effect [2]. For example, concomitant treatment with rifampin reduced both plasma concentrations of sulfonylureas and their blood glucose lowering effect possibly due to induction of CYP2C9 [130,131]. Induction of P-glycoprotein by rifampin might have also contributed to the reduced concentrations and effectiveness of the drugs [130,131]. In addition, although induction is generally considered less of a risk compared to inhibition, the increased metabolism of the victim drug can result in the formation of toxic metabolites [2].

Unlike CYP inhibition, which can rapidly affect drug metabolism by blocking the enzyme activity, CYP enzyme induction is a slower process [132]. CYP induction involves the upregulation of the enzyme biosynthesis, which takes time to reach a new steady-state level [132]. CYP induction by exogenous compounds is mainly mediated by three receptor-dependent mechanisms that can activate transcription of the genes that encode CYP families 1–3 [133,134]. These receptors include aryl hydrocarbon receptor (AhR), which belongs to the basic-helix-loop-helix-Per-Arnt-Sim (bHLH-PAS) family, pregnane X receptor (PXR), and constitutive androstane receptor (CAR) [2,133,135]. In the absence of ligands, these receptors exist in a latent state in the cytoplasm bound to heat shock protein 90 (Hsp90). When bound to ligands, the receptors undergo a conformational change in the ligand-binding domain, leading to the release of Hsp90, activation, and translocation to the nucleus for transcription (described in more details below).

In addition to AhR-, PXR-, and CAR-dependent mechanisms, other mechanisms have been implicated in CYP induction. For example, direct and indirect glucocorticoid receptor-mediated CYP induction have been reported in various studies [34,136,137]. Moreover, some CYPs are regulated at the level of mRNA and protein stabilization [138,139,140]. These transcriptional and post-transcriptional mechanisms are well reviewed in [141,142,143].

#### 4.1.1. Aryl Hydrocarbon Receptor

Extensive studies in the past two decades have identified many natural dietary and endogenous ligands of AhR and unmasked various physiological functions in normal development and homeostasis [144,145,146]. However, most AhR ligands are toxic xenobiotics, including halogenated aromatic hydrocarbons and polycyclic aromatic hydrocarbons [134,145]. In addition, AhR binds with several pharmaceuticals that are used clinically for multiple disorders [147]. Ligand binding mediates translocation of AhR to the nucleus where it heterodimerizes with another bHLH-PAS protein AhR nuclear translocator (ARNT) [132,134]. The AhR-ARNT complex associates with the xenobiotic response elements (XRE) in the promoter of target genes and recruits coactivators, such as SRC-1, CBP/p300, and NCOA-2 [134].

AhR is broadly distributed in human tissues, with its highest expression in the placenta, lung, heart, pancreas, and liver [148]. AhR primarily regulates *CYP1A1* and *CYP1B1* in extrahepatic tissues, while it targets *CYP1A2* in the liver [149]. Generally, CYP1A1 displays much higher sensitivity to AhR inducers compared to CYP1A2 and CYP1B1, which may be attributed to the presence of multiple XRE sites in *CYP1A1* [132,134]. In addition to CYP1, AhR has been shown to regulate some members in the CYP2 family [150,151,152]. However, it is important to note that there are species-specific differences. For example, Cyp2a5 is regulated by AhR in mice, though there is currently no equivalent evidence supporting the regulation of its human ortholog CYP2A6 [150].

#### 4.1.2. Nuclear Receptors

NRs are ligand-regulated transcription factors that regulate many physiological processes such as metabolism, inflammation, reproduction, and cell growth. A number of NRs have been identified as playing a role in regulating the expression of CYPs in response to xenobiotics. PXR and CAR are two well-studied NRs that serve as central regulators of CYPs. Both PXR and CAR have a ligand-binding domain that can bind a wide variety of ligands. Particularly, the ligand-binding domain of PXR is very large (1200–1600 Å^3^), highly hydrophobic, and flexible [153,154,155]. This unique structural feature enables it to bind a wide variety of molecules with varying size and structure, and allows a single ligand to engage in multiple orientations. Although the ligand-binding domain of CAR is hydrophobic, it is smaller (~600 Å^3^) and less flexible compared to that of PXR [155], making the receptor activated with a smaller number of ligands.

The DNA-binding domain and ligand-binding domain of NRs exhibit a high degree of homology and conservation across the species. However, the ligand-binding domain of PXR is significantly different across the species. Human, mouse, rat, and rabbit PXR orthologues show 75–82% amino acid sequence identity in their ligand-binding domain [156,157,158]. Similarly, the human and rodent CAR share ~70% sequence identity in their ligand-binding domain [159]. The poor homology among species is thought to lead to the marked species variation in ligand preferences and the induction of CYPs [156,157,158]. For example, rifampicin exhibited minimal activity when interacting with mouse PXR, but was a highly efficient activator of human PXR [160]. Conversely, activation of mouse PXR by pregnenolone 16α-carbonitrile was approximately three-fold higher compared to human PXR activation. Similar to PXR, studies demonstrated variation in ligand preferences of CAR across the species. 6-(4-chlorophenyl)imidazo[2,1-b][1,3]thiazole-5-carbaldehyde-O-(3,4-dichlorobenzyl)oxime (CITCO) activates CAR in humans but not in mice [161]. In contrast, 1,4-bis-[2-(3,5-dichloropyridyloxy)]benzene,3,3′,5,5′-tetrachloro-1,4-bis(pyridyloxy)benzene (TCPOBOP) is a stronger inducer for mouse CAR than human CAR [159,162]. To overcome the species differences in ligand recognition for translation of in vivo results from animal models to humans, various humanized mouse models for PXR, CAR, and CYPs have been developed [163,164,165,166].

In humans, both PXR and CAR are predominantly expressed in the liver with minimal levels detected in other tissues [142,167,168]. PXR regulates expression of several members in the subfamilies of *CYP2A*, *CYP2B*, *CYP2C*, and *CYP3A* [132,134,142]. Among these members, regulation of *CYP3A4* by PXR has been studied extensively [169,170,171]. Following translocation to the nucleus, PXR dimerizes with another NR retinoid X receptor (RXR) and then the PXR-RXR complex binds to AG(G/T) TCA-like direct repeats separated by three or four bases (DR3 and DR4), along with an everted repeat separated by 6 bases (ER6) [170]. These binding interactions occur at several distinct sites on the *CYP3A4* gene, including ER6 in the proximal promoter, DR3 in the xenobiotic-response element module (XREM), ER6 in a distal enhancer module, and the DR4 motif [169,170,171]. The DNA-bound PXR-RXR complex then activates transcription through recruitment of multiple coactivators including SRC-1, p300, and PGC-1 [134]. Similar to PXR, CAR forms a heterodimer complex with RXR in the nucleus. CAR-RXR complex regulates expression of the *CYP2B* gene by binding to DR4 motifs in the XREM [172]. Several coactivators of CAR, such as SRC-1, PGC-1, and ASC-2, have been identified [173].

#### 4.1.3. Crosstalk between Receptors

The significance of the interplay between receptor signaling pathways in CYP expression has been widely studied. This coregulation by receptors can occur at three levels: (i) sharing common ligands, (ii) receptor-receptor interactions, and (iii) sharing DNA-binding elements [134]. Notably, PXR and CAR have been associated with their involvement in mediating the effects of xenobiotics on the expression of *CYP2B6* and *CYP3A4*. For example, rifampicin, a potent PXR ligand and *CYP3A4* inducer, has been shown to induce *CYP2B6* in primary human hepatocytes. This induction is associated with the binding of PXR to the response elements in the *CYP2B6* gene [174,175]. In healthy volunteers taking a single dose of efavirenz, rifampicin enhanced CYP2B6-mediated efavirenz 8-hydroxylation and decreased the AUC of efavirenz. [176]. However, in vitro experiments demonstrated that CYP3A4 is more sensitive to induction by rifampicin than CYP2B6 [177]. Conversely, CAR can induce *CYP3A4* expression by binding to the distal XREM and promoter proximal regions of the gene [169]. In addition, Fahmi et al. demonstrated co-induction of CYP3A4 with various CYP2B6 inducers [177].

A recent study showed that PXR also interacts with AhR to regulate CYP expression [178]. The expression of PXR mRNA was diminished when primary hepatocytes were exposed to the AhR ligand, 2,3,7,8-tetrachlorodibenzo-p-dioxin (TCDD). Furthermore, rifampicin-induced *CYP3A4* expression was reduced in the presence of TCDD, suggesting a negative regulatory effect of AhR on *CYP3A4* expression. However, the mechanisms by which AhR activation regulates the expression of PXR and CYP3A4 are still not clear.

### 4.2. Methods for Assessing CYP Induction

#### 4.2.1. Primary Human Hepatocytes

Cultured primary human hepatocytes express all the relevant hepatic metabolic enzymes, transporters, and their regulators [134]. Among the various in vitro or cell-based approaches, the use of primary hepatocytes is recommended by the FDA and EMA as these cells provide results that are closest to in vivo studies [2,134]. During drug development, initial experiments can be conducted to evaluate CYP1A2, CYP2B6, and CYP3A4/5 [5]. If CYP3A4/5 induction is positive, however, the potential of CYP2C induction by the test drug should be evaluated in the follow-up studies since both CYP3A4/5 and CYP2C enzymes are induced by PXR activation [2]. The incubation duration for inducers typically ranges from 48 to 72 h, which allows the complete induction of enzyme. During the incubation period, the inducer is added daily by replacing the medium containing the drug. Commonly used CYP inducers are presented in Table 1. The standard endpoints involve the measurement of mRNA levels and/or enzyme activity.

In general, the fold change in mRNA and enzyme activity measurements are thought to be consistent with each other [129]. However, a major challenge with enzyme activity assays is the potential for mixed outcomes when the test compound acts as both inducer and inhibitor [5]. For example, primary human hepatocytes exposed to DPC 681, a selective human immunodeficiency virus (HIV) protease inhibitor, had significant increases in CYP3A4 mRNA and protein levels [179,180]. However, CYP3A4 metabolic activity did not increase due to the inhibition effect of DPC 681. To address this challenge, assessment of induction using transcriptional analysis through mRNA measurement is recommended by the FDA [5]. The test compound can be considered as an inducer when there is a dose-dependent increase in mRNA expression that exceeds two-fold compared to the vehicle control or if the mRNA reaches at least 20% of the level observed in the positive control treated with known inducers. In addition, it is important to validate the system by generating full dose response curves of the positive controls to show that CYP enzymes are functional and inducible. Examples of positive controls include 20 μM of omeprazole, 1 mM of phenobarbital, and 10 μM of rifampicin for CYP1A2, CYP2B6, and CYP3A4, respectively [2].

As follow-up to any dose-dependent positive response, definitive studies can be conducted to determine the E_max_ (maximum induction effect) and EC_50_ (concentration causing half-maximal effect). During drug development, these studies must be conducted in primary human hepatocytes from at least three individual donors to address variability in individual responses to inducers [2,3]. Multiple concentrations of the test compound, usually 4 to 8, covering up to one order of magnitude (10×) over the C_max_ are typically used. Once the E_max_ and EC_50_ are determined, several approaches can be used for further prediction of enzyme induction [181,182]. The basic kinetic model involves a direct comparison of values determined from the in vitro E_max_, EC_50_, and plasma concentration of test compounds with the uniform threshold of DDI risk [181]. In this approach, a positive induction is determined by the formula:R3=1[1 + d × ((Emax × 10 × Imax,u)(EC50 + 10 × Imax,u))]
where I_max,u_ is the maximal unbound inducer plasma concentration at steady state, and d is the induction scaling factor [5]. Correlation methods determine the magnitude of the inducible effect according to a calibration curve of relative induction score (RIS score, (E_max_ × [I]/(EC_50_ + [I]))) with known inducers or I_max,u_/EC_50_ with known inducers and non-inducers. The advantages and disadvantages of each approach are described in [181].

Using primary human hepatocytes in induction assays has several limitations, including donor variability, limited availability, short viability, and batch-to-batch variation [132,183,184]. The major concern with 2D monolayer primary hepatocyte cultures is the rapid de-differentiation after plating and loss of hepatic functions, including drug metabolizing enzyme activity [183,185]. Several cell models have been developed to prevent or ameliorate the de-differentiation [186,187,188]. Overlaying 2D cultures with a thin layer of extracellular matrix (ECM) improves cellular phenotypes and polarization [186]. However, a decline in CYP3A activity, along with a decline in albumin secretion and urea production, was demonstrated after two weeks in culture [187]. The 3D spheroid primary hepatocytes have been suggested as an emerging tool to study drug metabolism as they maintain in vivo hepatic characteristics and stably express CYPs for several weeks in culture [188]. Furthermore, a recent study by Järvinen et al. demonstrated that mRNA and protein levels of CYPs are induced by different inducers in spheroids [189]. A major limitation associated with using spheroids is that not all batches of primary hepatocytes have the capacity to develop into spheroids [189].

#### 4.2.2. Immortalized Hepatocytes

Immortalized hepatic cell models offer several advantages for CYP induction studies over primary hepatocytes: easy accessibility, low variability, highly reproducible results in response to inducers, and high availability. Simian virus 40 immortalized human hepatic Fa2N-4 cells express CYP1A2, CYP2C9, and CYP3A4 protein, and the mRNA expression and activity of these enzymes are inducible in response to prototypical inducers [190]. In addition, Ripp et al. demonstrated a comparable increase in mRNA levels of *CYP3A4* using 24 compounds in Fa2N-4 cells and primary human hepatocytes [191]. However, CAR-selective activators CITCO and artemisinin did not induce CYP3A4 and CYP2B6 mRNA levels, possibly due to very low expression of CAR in Fa2N-4 cells [192]. Furthermore, rifampicin resulted in a 10-fold higher EC_50_ in Fa2N-4 cells than cryopreserved human hepatocytes, possibly due to low expression of the hepatic uptake transporters organic anion-transporting polypeptides OATP1B1 and OATP1B3 [192].

HepaRG is a human hepatoma cell line that expresses the major CYPs and their regulators including CAR at levels similar to those observed in freshly isolated hepatocytes after differentiation with DMSO [183,193]. HepaRG cells respond to prototypical inducers of CYP1A1, CYP1A2, CYP2B6, CYP2C8, CYP2C9, CYP2C19, and CYP3A4 at both mRNA and enzyme activity levels [193,194,195,196]. A strong correlation between EC_50_ for CYP3A4 induction following rifampicin treatment has been demonstrated in HepaRG cells and primary human hepatocytes [196]. Furthermore, studies showed that the results from HepaRG cells can be used to predict the in vivo induction effect of drugs using different calculation models [195,197,198]. These indicate that HepaRG cells are an excellent surrogate for predicting CYP3A induction potential by drugs in primary hepatocytes and in vivo.

#### 4.2.3. High Throughput Assays

PXR receptor assays are widely used high throughput assays due to their importance in DDI studies. The cell-free ligand binding assays typically involve quantification of the competition between the test compound and a radiolabeled ligand for receptor binding in genetically expressed and isolated receptor preparations. However, there are instances in which substantial ligand-receptor binding fails to trigger transactivation, resulting in false positives. On the other hand, cell-based transactivation assays are more accurate with less false positives and better correlate with human DDIs [3]. In the transactivation assays, two expression vectors, full length human PXR and a variation of the target promoter coupled to a reporter such as luciferase, are transfected into host cells [3,183]. Host cells are then treated with the test compound, and the reporter gene activity is assayed. Increased reporter gene activity is an indication of induction, and a dose-response curve is generated to determine E_max_ and EC_50_ [3,133].

These assays offer several advantages over using cultured cells, including their simplicity to conduct, high capacity, and cost-effectiveness [134,183]. Consequently, they can be particularly beneficial in the early phase of drug discovery [183]. However, one major limitation is their capacity to assess only one receptor-mediated induction pathway at a time, as crosstalk between receptors can activate the same target genes [3]. In addition, the cell-free ligand binding assays are unable to account for other mechanisms such as the impact of cell membrane, which may restrict cellular uptake of drugs [3,134].

## 5. Additional Considerations for Drug Interactions

As noted above, CYPs are involved in the metabolism of approximately 70–80% of small molecule drugs in clinical use [7,199]. For this reason, CYPs are frequently involved in DDIs because co-administered drugs are often substrates, inducers, and/or inhibitors of CYPs. Human CYP enzymes of the 1, 2, and 3 families play a major role in drug metabolism. These enzymes include CYP1A2, CYP2B6, CYP2C9, CYP2C8, CYP2C19, CYP2D6, CYP3A4, and CYP3A5. CYP3A4 is the most abundant CYP in adult human liver and intestine, and this enzyme is involved in the metabolism of approximately 50% of small molecule drugs in clinical use or in development [199,200]. The large flexible active site of CYP3A4 and its major role in drug metabolism make it particularly susceptible to DDIs [201]. Further, more than one ligand can occupy the CYP3A4 active site at one time [201]. In addition to DDIs, drug interactions can involve ingested or inhaled xenobiotics, such as ethanol (CYP2E1 inducer) and tobacco smoke (CYP1A1 and CYP1A2 inducer); herbal supplements, such as St. John’s Wort (CYP3A4 inducer); and food, such as components of grapefruit juice (CYP3A4 inhibitor) [199]. Beyond the drug-metabolizing CYPs, other CYP enzymes are involved in the metabolism of steroids (e.g., CYP7A1), fatty acids (e.g., CYP4A11), and vitamins (e.g., CYP26A1). Still other CYP enzymes currently have no known substrates (i.e., “orphan” CYPs) [199].

Drug interaction studies can be complicated by the fact that some drugs can induce more than one CYP enzyme, and some drugs can inhibit more than one CYP enzyme. Some examples are provided below. First, as noted above, rifampicin is an agonist of the nuclear receptor PXR. PXR regulates the expression of multiple CYP enzymes and drug transporters [202]. CYP enzymes induced by rifampicin include CYP2B6, CYP2C8, CYP2C9, CYP2C19, and CYP3A4 [82]. Rifampicin also induces P-glycoprotein [203]. In addition, rifampicin can inhibit the transporters organic anion transporting polypeptide (OATP) 1B1 and OATP1B3. DDIs involving rifampicin are an important clinical concern [203]. Second, azole antifungals are a common cause of drug interactions due to CYP inhibition. The selectivity of CYP inhibition depends on the concentration of inhibitor [68]. For example, ketoconazole is a potent inhibitor of CYP3A4 and CYP3A5 at nanomolar concentrations in vitro [204]. At higher concentrations, ketoconazole can inhibit other CYP enzymes. In addition, some drugs inhibit multiple CYP enzymes at therapeutically relevant concentrations. For example, fluconazole is a strong inhibitor of CYP2C19 and a moderate inhibitor of CYP2C9 and CYP3A4 [82]. Therefore, DDIs with fluconazole are a concern in clinical practice [205]. Additionally, a compound can have different effects on the activity of CYP enzymes. For example, α-naphthoflavone is an inhibitor of CYP1A2 and an activator of CYP3A4 [206].

Atypical (non-Michaelis–Menten) kinetic profiles have been reported for CYP enzymes, which may be characterized by sigmoidal autoactivation, substrate inhibition, or biphasic kinetic profiles [207]. In addition, experimental conditions, such as nonspecific binding of substrate to microsomal preparations, may result in the appearance of “atypical” kinetic profiles, which are artifacts of the experimental set-up [207]. The impact of in vitro observed atypical kinetic profiles on IVIVE and DDI predictions requires further investigation [208].

## 6. Conclusions

Multiple drug therapy is becoming increasingly more common to enhance therapeutic efficacy. Many drugs undergo hepatic metabolism by CYP enzymes for clearance from the body. Over the years, significant advancements have been achieved on in vitro CYP assays and predictions for DDIs. The information gathered from in vitro studies not only helps in the identification of risks associated with multiple drug therapy but also provides insights into the fundamental mechanisms driving these interactions. While in vitro DDI assessment has advanced our understanding, challenges persist, including the need for standardized protocols, consideration of interindividual variation, and the incorporation of physiologically relevant conditions. Future research should focus on refining methodologies, fostering the development of more predictive models that closely mirror the complexities of in vivo interactions.

## Figures and Tables

**Figure 1 biomolecules-14-00099-f001:**
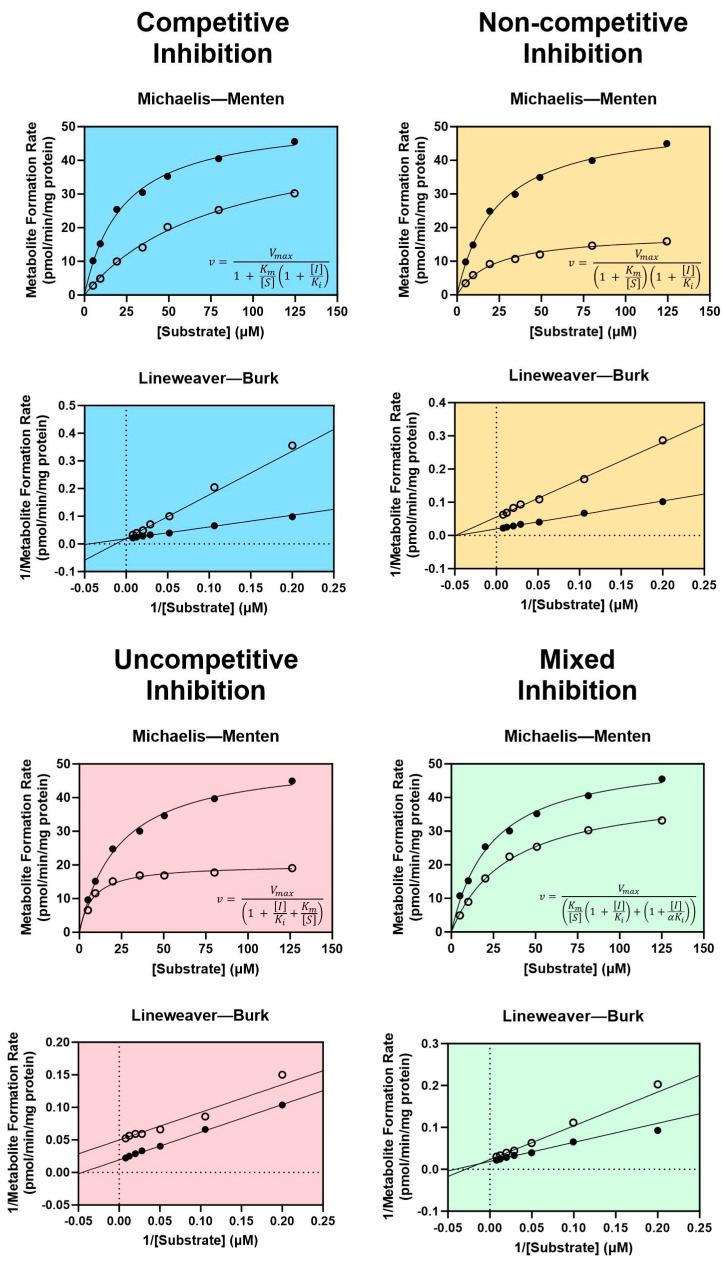
Enzyme kinetics of reversible inhibition. Characteristic Michaelis–Menten curves and equations are shown with their corresponding Lineweaver–Burk plots for competitive inhibition (blue), non-competitive inhibition (yellow), uncompetitive inhibition (pink), and mixed competitive/non-competitive inhibition (green). Filled circles represent metabolite formation rates measured without an inhibitor, and open circles represent metabolite formation rates measured with an inhibitor. *v* = the rate of metabolite formation, V_max_ = the maximal rate of metabolite formation, K_m_ = the substrate concentration at the half-maximal rate of metabolite formation, [I] = inhibitor concentration, [S] = probe substrate concentration, K_i_ = the equilibrium dissociation constant for the enzyme-inhibitor complex, αK_i_ = the equilibrium dissociation constant for the enzyme–substrate–inhibitor complex. Dashed lines in the Lineweaver–Burk plots indicate x = 0 (vertical dashed line) and y = 0 (horizontal dashed line). Figure adapted from Ring et al. using GraphPad Prism version 10.1.2 software [102].

**Figure 2 biomolecules-14-00099-f002:**
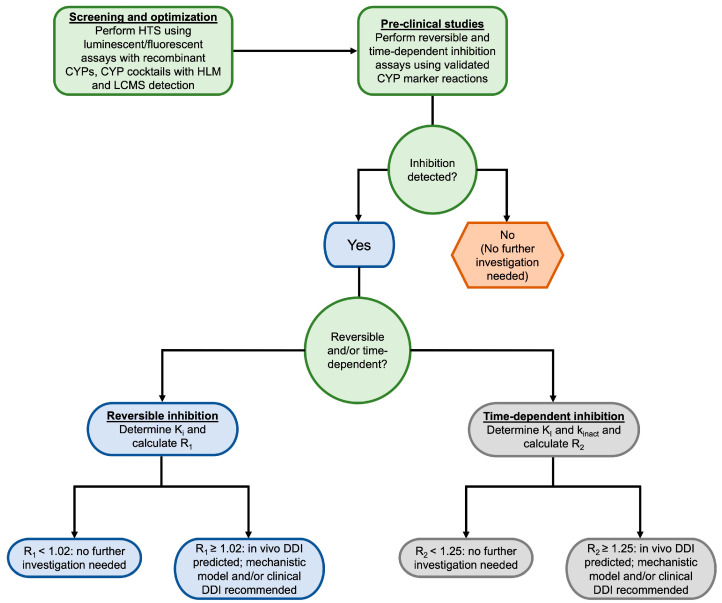
Decision tree for studying CYP inhibition in vitro. Decision nodes are based on the 2020 FDA Guidance for Industry for in vitro drug interactions [5]. HTS = high throughput screening, CYP = cytochrome P450, HLM = human liver microsome, LCMS = liquid chromatography-mass spectrometry, DDI = drug-drug interaction.

**Table 1 biomolecules-14-00099-t001:** Examples of in vitro substrate marker reactions, inhibitors, and inducers for CYP-mediated drug metabolism.

CYP Enzyme	Substrate Marker Reactions	Inhibitors	Inducers
1A2	Caffeine 3-*N*-demethylation [9,10,11] 7-ethoxyresorufin *O*-deethylation [12,13,14] Phenacetin *O*-deethylation [15,16,17,18]	Amiodarone [19,20]	Omeprazole [15,21]
		Cimetidine [22,23,24]	
		Furafylline [15,18,25,26]	
		α-Naphthoflavone [27,28]	
2B6	Bupropion hydroxylation [17,18,29] Efavirenz hydroxylation [30]	Clopidogrel [31,32]	Phenobarbital [33,34,35,36]
		Ticlopidine [31,32]	
		Sertraline [32,37]	
		Thiotepa [31,38]	
		2-phenyl-2-(1-piperidyl)propane [32,39]	
2C8	Amodiaquine *N*-deethylation [18,40,41] Paclitaxel 6α-hydroxylation [40,42]	Gemfibrozil [43]	Rifampicin [34,36,44]
		Montelukast [45]	
		Phenelzine [46]	
2C9	Diclofenac 4′-hydroxylation [18,47,48,49]	Sulfaphenazole [18,49,50,51]	Rifampicin [34,36,44,52]
	*S*-warfarin 7-hydroxylation [48]	Tienilic acid [17,53]	
2C19	*S*-mephenytoin 4′-hydroxylation [54]	*N*-3-benzylnirvanol [55]	Rifampicin [36]
		Ticlopidine [51,56]	
		Loratadine [57]	
		Nootkatone [58]	
2D6	Bufuralol 1′-hydroxylation [49,59]	Quinidine [18,49,60,61]	
	Dextromethorpan *O*-demethylation [18,62]	Paroxetine [61,63]	
3A4/5	Midazolam 1′-hydroxylation [17,49,64,65,66]	Ketoconazole [17,49,50,64,65,66]	Rifampicin [15,34,36,67]
	Testosterone 6β-hydroxylation [17]	CYP3Cide (3A4 specific) [65,66]	

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
