# Peer review of "A Review of CYP-Mediated Drug Interactions: Mechanisms and In Vitro Drug-Drug Interaction Assessment"

_biomolecules, 2024, doi:10.3390/biom14010099_

Round 1
Reviewer 1 Report
Comments and Suggestions for Authors
Author reviewed the cytochrome p450 mediated drug interaction which can influence the net activity of the enzymes. Those effect may be inhibitory or indulgent based on the properties of interaction. Drug-drug interaction is a challenge for current days medicinal strategy and may lead to personalized medicine progression. Involvement of CYPs for this kind of study will add valuable information. In my opinion this review is great to focus drug-drug interaction involving CYPs.
1) Why CYPs are mainly involved in drug -drug interaction? Is there any protein structure or active site relevance for these enzymes?
2) Out of 57 CYPs in humans, few are involved in drug metabolism, those are mainly located in liver. What about other CYPs?
3) Can similar drugs affect different enzymes or proteins? In more clear term whether drugs are lacking specificity?
4) Drug-drug interaction can be affected by other chemicals like alcohol.
5) Hydrophobic drugs can interact themselves, especially in aqueous environment. Is there a way to identify those self-interaction and is there any report to affect Kd and Km of the enzyme?
6) How to explain non classical inhibitory plot for substrate binding?
Comments on the Quality of English LanguageEnglish is good
Reviewer 2 Report
Comments and Suggestions for Authors
The concomitant use of multiple drugs—often referred to as polypharmacy—is becoming increasingly more common in clinical practice. Drug-drug interactions (DDIs) involving cytochrome P450 (CYP) enzymes are an ongoing challenge associated with polypharmacy because they can lead to alteration in the exposure of the victim drug, potentially resulting in diminished benefits and/or heightened risks. Over the years, significant advancements have been achieved with in vitro CYP assays and predictions for DDIs. This article reviews current understanding of mechanisms of inhibition and induction of CYP enzymes and in vitro approaches to assess CYP-mediated DDIs.
The reviewer offers the following comments, by section of the manuscript, for the authors’ consideration.
Abstract
In the second sentence of the Abstract, the authors stated that DDIs can lead to alteration in the exposure of the victim drug, raising safety concerns. In addition to safety concerns, as the authors can appreciate, alteration in the exposure of the victim drug can also raise effectiveness concerns. A prime example is a DDI involving a prodrug as the victim drug. For instance, a CYP2C19 inhibitor (i.e., perpetrator drug) can alter (reduce) the biotransformation of clopidogrel (i.e., victim drug) to its active metabolite(s) responsible for the antiplatelet effect of the parent compound. Another example is a DDI involving a CYP450 inducer, whereby enzymatic induction by a perpetrator drug can alter (enhance) the biotransformation of the victim drug, leading to reduced drug concentrations of the latter in the body. As such, it may behoove the authors to mention that altered effectiveness also is a potential concern with DDIs.
Reaction Phenotyping
In the Additional Methodologies subsection (page 6), the authors described another study that compared various machine learning models and predicted in vivo human IV clearance from in vitro data (lines 219-221). In this sentence, what does the abbreviation “IV” in “human IV clearance” indicate (e.g., in vivo, intravenous)? Please define this abbreviation for clarity.
CYP Inhibition
Regarding abbreviations used for figures, it is generally preferred to define each abbreviation used—in the figure caption—even if the abbreviation has already been defined in the manuscript text. As an example, the abbreviation “HTS” is the only abbreviation defined in Figure 2 (page 11), yet several other abbreviations are used in this figure but have not been defined in the figure caption. Please refer to the journal requirements for authors to determine whether all abbreviations (versus only newly used abbreviations) should be defined in each figure.
CYP Induction
In the first paragraph of the Mechanisms of CYP Induction subsection, the authors stated that “concomitant treatment with rifampin reduced plasma concentrations of sulfonylureas and a blood glucose lowering effect occurred due to induction of CYP2C9” (page 14, lines 489-491). First, did the authors intend to state that the blood-glucose lowering effect of sulfonylureas was reduced during concomitant administration with rifampin? Assuming this is the point the authors intended to make, a rewording of this sentence might be beneficial (e.g., “…reduced both plasma concentrations of sulfonylureas and their blood-glucose lowering effect due to…). Second, the authors might consider mentioning that induction of P-glycoprotein by rifampin may have also contributed to the observed results reported by the cited studies.
